# The Hypotension Period after Initiation of Appropriate Antimicrobial Administration Is Crucial for Survival of Bacteremia Patients Initially Experiencing Severe Sepsis and Septic Shock

**DOI:** 10.3390/jcm9082617

**Published:** 2020-08-12

**Authors:** Ching-Chi Lee, Chao-Yung Yang, Bo-An Su, Chih-Chia Hsieh, Ming-Yuan Hong, Chung-Hsun Lee, Wen-Chien Ko

**Affiliations:** 1Clinical Medicine Research Center, National Cheng Kung University Hospital, College of Medicine, National Cheng Kung University, Tainan 70403, Taiwan; chichingbm85@gmail.com; 2Department of Internal Medicine, National Cheng Kung University Hospital, College of Medicine, National Cheng Kung University, Tainan 70403, Taiwan; 3Department of Emergency Medicine, National Cheng Kung University Hospital, College of Medicine, National Cheng Kung University, Tainan 70403, Taiwan; chao.youg@gmail.com (C.-Y.Y.); hsiehchihchia@gmail.com (C.-C.H.); myuan@mail2000.com.tw (M.-Y.H.); chlee82er@yahoo.com.tw (C.-H.L.); 4Division of Infectious Disease, Department of Internal Medicine, Chi Mei Medical Center, Tainan 71004, Taiwan; suboan0421@gmail.com; 5Department of Medicine, National Cheng Kung University Medical College, Tainan 70101, Taiwan

**Keywords:** bacteremia, empirical, antibiotic, vasopressor agents, severe sepsis, septic shock

## Abstract

Bacteremia is linked to substantial morbidity and medical costs. However, the association between the timing of achieving hemodynamic stability and clinical outcomes remains undetermined. Of the multicenter cohort consisted of 888 adults with community-onset bacteremia initially complicated with severe sepsis and septic shock in the emergency department (ED), a positive linear-by-linear association (*γ* = 0.839, *p* < 0.001) of the time-to-appropriate antibiotic (TtAa) and the hypotension period after appropriate antimicrobial therapy (AAT) was exhibited, and a positive trend of the hypotension period after AAT administration in the 15-day (*γ* = 0.957, *p* = 0.003) or 30-day crude (*γ* = 0.975, *p* = 0.001) mortality rate was evidenced. Moreover, for every hour delay of the TtAa, 30-day survival dropped an average of 0.8% (adjusted odds ratio [AOR], 1.008; *p* < 0.001); and each additional hour of the hypotension period following AAT initiation notably resulted in with an average 1.1% increase (AOR, 1.011; *p* < 0.001) in the 30-day crude mortality rate, after adjusting all independent determinants of 30-day mortality recognized by the multivariate regression model. Conclusively, for bacteremia patients initially experiencing severe sepsis and septic shock, prompt AAT administration might shorten the hypotension period to achieve favourable prognoses.

## 1. Introduction

Bacteremia is a severe life-threatening condition that is associated with substantial morbidity and mortality [1]. As its initial presentation of severe sepsis or septic shock, a short-term case fatality rate can reach up to 50% [2], and the advantage of appropriate administration of empirical antimicrobials has been evidenced [3,4,5]. Accordingly, to improve outcomes of patients initially experiencing severe sepsis and septic shock, the identification of infection sites amenable to undergo source control and prompt administration of effective antimicrobial therapy were emphasized by the Surviving Sepsis Campaign (SSC) [6]. Additionally, other recommendations to rapidly achieve hemodynamic stability, in terms of the fluid challenge technique, appropriate vasopressor medications, adequate corticosteroid dosing, and indication of blood transfusion, were comprehensively recognized in the SSC guideline [6]. However, the beneficial effect of rapid hemodynamic stability after administration of appropriate antimicrobial therapy (AAT) on the survival of bacteremia patients initially with the critical illness has not been previously identified. We hypothesized a shorter hypotension period after AAT administration resulted in better prognoses of bacteremic patients initially presenting with severe sepsis and septic shock. Accordingly, for adults having community-onset bacteremia and initially experiencing severe sepsis and septic shock, the aims of the present study included: (i) studying the linear-by-linear association of the AAT timing and the hypotension period after AAT administration; (ii) investigating the correlation of this hypotension period after AAT administration and mortality rates; and (iii) respectively, identifying the independent impact of the AAT timing and the hypotension period after AAT administration on patient fatality.

## 2. Methods

### 2.1. Study Design and Population

This four-year, multicenter cohort study was retrospectively conducted during the period between January 2016 and December 2019 in the emergency departments (EDs) of three hospitals in southern Taiwan. One study hospital is a university-affiliated medical center approximately with 1400 beds and two are teaching hospitals respectively consisted of 460 and 380 beds. The patients (aged ≥ 18 years) with community-onset bacteremia and initially presenting with severe sepsis and septic shock in the ED were included. All patients were managed by board-certified ED physicians during the ED stay and further cared by board-certified intensive care unit (ICU) physicians, if patients had admitted to ICUs through the ED. All resuscitations were directed or operated by physicians and nurses using the guidelines issued by American Heart Association (AHA)/advanced cardiac life support (ACLS). The study was approved by the institutional review board of each participating hospital (B-ER-100-182, 5th ed, SLH 9919-108-006, and SLH 9919-108-009) and the waiver of the requirement for informed consent was obtained under the regulations of each hospital due to the retrospective nature of the present study.

Patients with blood culture sampling in the ED were retrieved from a computer database. For patients with microbial growth in blood cultures, their medical records were initially reviewed, including patient demographics, vital signs, the presence of systemic inflammatory response syndromes or sepsis-related syndromes (i.e., severe sepsis and septic shock) immediately after each patient’s ED visit, microorganism types, and bacteremia sources. Only the first episode of each patient, if multiple bacteremia episodes were reported, was included. After exclusion of patients with contaminated blood cultures, those with fungemia or mycobacteremia, those not initially experiencing severe sepsis and septic shock, and those with bacteremia diagnosed before the ED visit, the study finally enrolled only patients having community-onset bacteremia complicated with severe sepsis and septic shock.

### 2.2. Data Collection

By reviewing the medical chart (electronic and paper records), a predetermined form was adapted to collect patient demographics and clinical characteristics, in terms of age, gender, transfer sources, bacteremia severity (a Pitt bacteremia score), initial syndromes (e.g., septic shock and severe sepsis), timing and types of vasopressor agents, comorbidity severity (the McCabe classification), comorbidities, image studies, and laboratory data. Further hospitalization through the ED, such as types and durations of antimicrobial administration, microbiological results, imaging studies, sources of bacteremia, dates and types of radiologic or surgical interventions, lengths of hospitalization, and clinical outcomes were also recognized. All the clinical data were randomly retrieved by two authors, and the two authors inspected the medical records together if any discrepancies were found. The study endpoint was crude mortality within 30 days after ED arrival (i.e., bacteremia onset), and patients were excluded if they had incomplete clinical information or the uncertain fatality when collection.

### 2.3. Microbiological Methods

Blood cultures were incubated in a BACTEC 9240 instrument (Becton Dickinson Diagnostic Systems, Sparks, MD, USA) for 5 days at 35 °C. Bacterial species was identified by means of the Vitek 2 system (bioMérieux, Durham, NC, USA). All bacteremic isolates in the study period were prospectively collected for the antimicrobial susceptibility test determined by the broth microdilution method in accordance with the contemporary Clinical and Laboratory Standards Institute (CLSI) standard [7]. The tested antibiotics for Gram-negative aerobes included ampicillin/sulbactam, piperacillin/tazobactam, cefazolin, cefuroxime, cefotaxime, ceftazidime, cefepime, ertapenem, imipenem, moxifloxacin, and levofloxacin. For *Streptococcus* species and *Enterococcus* species, the tested drug was penicillin and ampicillin, respectively. Cefoxitin was tested for *Staphylococcus aureus* to identify the methicillin-resistant isolate. To ensure the AAT timing in each bacteremia episode, if a patient empirically treated by an antibiotic which was not included in the tested antimicrobials offered by the original study design, the susceptibility to the indicated agent was retrospectively measured.

### 2.4. Definitions

Bacteremia was defined as bacterial growth of blood cultures drawn from central or peripheral venipuncture, after the exclusion of contaminant sampling. Blood cultures sampling with potentially contaminating pathogens, such as coagulase-negative staphylococci, micrococci, *Bacillus* species, *Propionibacterium* species, and Gram-positive bacilli, were considered to be contaminated based on the previous criteria [8]. Community-onset bacteremia indicates that the onset place of the bacteremia episode is the community [9,10], which includes healthcare- and community-associated bacteremia. The isolation of ≥ two microbial species from a single bacteremia episode was regarded as polymicrobial bacteremia.

As previously described [3,4], AAT administration was considered when all the following criteria were fulfilled: (i) the administrated antimicrobial was in vitro active against all the causative microorganisms isolated from blood cultures, based on the contemporary CLSI breakpoint [7]; (ii) the route and dosage of antimicrobials was administered as recommended in accordance with the *Sanford Guide to Antimicrobial Therapy 2016* [11]. The period (measured by hours) between the bacteremia onset (i.e., ED triage) and AAT administration was regarded as the time-to-appropriate antibiotic (TtAa) [3,4]. As the SSC recommendation [6], complicated bacteremia was defined by determination of whether that the bacteremia source is amenable to source control, such as the drainage of an abscess or obstructive tract, removal of a potentially infected device, debridement of infected necrotic tissue, and definitive control of sources with ongoing bacterial contamination. As previously described [12,13], sources of bacteremia and the appropriateness of specific percutaneous or surgical source control were determined by infectious diseases–trained physicians.

Hypotension was defined as a mean blood pressure of <65 mm Hg, a systolic blood pressure of <90 mm Hg, or a decrease in systolic pressure of 40 mm Hg from the baseline. The hypotension period after AAT administration was calculated by clinicians as the hour number from AAT initiation until discontinuation of all types of vasopressor agents, such as dopamine, dobutamine, and epinephrine, after the patients achieved hemodynamic stability. As previously reported [14], hemodynamic stability was defined as a mean arterial pressure of 75 mmHg and the lack of vasopressor administration. Comorbidities were defined as described previously [15]; and malignancies included hematological malignancies and solid tumors. The prognosis of preexisting diseases was assessed using a previous delineated classification system (McCabe classification) [16]. The sources of bacteremia followed the published definitions of the Centers for Disease Control and Prevention [17]. Crude mortality was used to define death from all causes.

A Pitt bacteremia score graded the severity of bacteremia onset using a previously established scoring system based on vital signs, the usage of vasopressor agents, mental status, the receipt of mechanical ventilation and recent cardiac arrest [18]. A patient initially presenting with a high Pitt bacteremia score (≥ 4 points) was defined as having the critical illness. As the traditional concept, severe sepsis was defined as the coexistence of sepsis and at least one of the following signs or symptoms of acute organ dysfunction or hypoperfusion: metabolic acidosis, arterial hypoxemia, oliguria, coagulopathy, or encephalopathy [19]; the presence of systemic inflammatory response syndromes and a systolic blood pressure no higher than 90 mmHg after an adequate crystalloid-fluid challenge or a blood lactate concentration of ≥ 4 mmol/L was regarded as septic shock [20].

### 2.5. Statistical Analysis

Statistical analyses were performed using the Statistical Package for the Social Science for Windows (Chicago, Illinois, USA), version 23.0. Categorical variables were expressed as numbers and percentages and compared using the Chi-square test or Fisher’s exact test. Continuous variables were presented as the median and interquartile range (IQR) and compared by the Student’s *t*-test. A linear-by-linear association between two continuous variables was analyzed using the *Pearson* correlation.

To study the independent impact of TtAa delay and the hypotension period after AAT administration on 30-day mortality, the variable of 30-day mortality recognized by the univariate analysis and two continuous variables, in terms of TtAa delay and the hypotension period, were together processed using a stepwise, backward logistic regression model. A *p* value < 0.05 was considered significant. 

## 3. Results

### 3.1. Demographics and Clinical Characteristics of the Study Cohort

The total 888 patients having community-onset bacteremia and initially experiencing severe sepsis and septic shock were enrolled based on the inclusion and exclusion criteria (Figure 1). The median (IQR) of patient ages was 71 (58–81) years, and 518 patients (58.3%) were male. Their median (IQR) lengths of ED stay and hospitalization were 16.3 (5.0–30.3) hours and 9 (5–19) days, respectively. Patients with complicated bacteremia accounted for 20.6% (183 patients) of the overall cohort. Most patients (672, 75.7%) had the critical illness (a Pitt bacteremia score ≥ 4) at ED arrival and thus a high 15-day (46.7%) and 30-day (53.2%) crude mortality rate was observed.

Of the total patients, the leading 10 comorbidities were hypertension (391, 44.0%), diabetes mellitus (329, 37.0%), malignancies (298, 33.6%), neurological disorders (293, 33.0%), chronic kidney diseases (181, 20.4%), liver cirrhosis (130, 14.6%), congestive heart failure (93, 10.5%), coronary artery diseases (93, 10.5%), urological diseases (92, 10.4%), and chronic obstructive pulmonary diseases (53, 6.0%). The most common source of bacteremia was pneumonia (326, 36.7%), followed by urinary tract infections (171, 19.3%), intra-abdominal infections (96, 10.8%), soft-tissue infections (76, 8.6%), biliary tract infections (53, 6.0%), primary bacteremia (47, 5.3%), liver abscess (35, 3.9%), vascular-line infections (32, 3.6%), bone and joint infections (25, 2.8%), and infective endocarditis (21, 2.4%).

### 3.2. Causative Microorganisms and Susceptibilities

Because 152 patients suffered polymicrobial bacteremia, the total 1084 causative microorganisms were captured. The leading 10 microorganisms included *Escherichia coli* (275, 25.4%), *Klebsiella pneumoniae* (210, 19.4%), *Streptococcus* species (141, 13.0%), *Staphylococcus aureus* (121, 11.2%), *Pseudomonas* species (51, 4.7%), *Proteus* species (43, 4.0%), *Enterococcus* species (35, 3.2%), *Enterobacter* species (25, 2.3%), *Vibrio* species (15, 1.4%), and *Salmonella* species (13, 1.2%). 

Methicillin-resistant *S. aureus* and ampicillin-susceptible enterococci accounted for 50.4% (61 isolates) of *S. aureus* and 91.4% (32) of *Enterococcus* species, respectively. Of 141 *Streptococcus* species, 93.6% (132 isolates) were susceptible to penicillin. Overall, cefazolin, cefuroxime, ampicillin/sulbactam, moxifloxacin, cefotaxime, levofloxacin, ceftazidime, ertapenem, cefepime, piperacillin/tazobactam, or imipenem was overall active against 58.6%, 72.3%, 78.2%, 80.3%, 82.1%, 85.6%, 86.6%, 92.3%, 93.0%, 94.6%, or 100%, respectively, of Gram-negative aerobes.

### 3.3. Association of the Time-to-Appropriate Antibiotic (TtAa) or Hypotension Periods and Mortality Rates

The median (IQR) of the TtAa and hypotension period after AAT initiation was 2 (1–11) and 80 (29–270) hours, respectively. A positive correlation (*γ* = 0.839, *p* < 0.001) of the TtAa and the hypotension period following AAT administration was exhibited (Figure 2). Furthermore, a positive trend of the hypotension period after AAT initiation in the 15-day (*γ* = 0.957, *p* = 0.003) and 30-day crude (*γ* = 0.975, *p* = 0.001) mortality rate was evidenced in Figure 3. 

### 3.4. Independent Impacts of the TtAa and Hypotension Period on 30-Day Mortality

The association of numerous clinical variables, including patient demographics, bacteremia severity at onset, bacteremia sources, severity of underlying diseases, major comorbidities, and causative pathogens, with 30-day mortality was examined using univariate analysis (Table 1). The following variables were positively associated with 30-day mortality: nursing-home residents, critically ill (a Pitt bacteremia score ≥ 4) patients at onset, inadequate source control during antimicrobial therapy, bacteremia pneumonia, causative microorganisms of *Streptococcus* species or *S. aureus*, fatal comorbidities (the McCabe classification), and underlying malignancies. Otherwise, several predictors, including bacteremia caused by urinary or biliary tract infections, and causative microorganisms of *E. coli,* were factors against 30-day crude mortality.

Under the multivariate regression model (Table 1), numerous independent predictors of 30-day mortality were identified: the critical illness (a Pitt bacteremia score ≥4) at onset, fatal comorbidities (the McCabe classification), bacteremia caused by urinary or biliary tract infections, the TtAa period, and the hypotension period after AAT initiation. Notably, each additional hour of TtAa delay and the increasing hypotension period was associated with an average increase in the 30-day mortality rate of 0.8% (adjusted odds ratio [AOR], 1.008; *p* < 0.001) and 1.1% (AOR, 1.011; *p* < 0.001), respectively.

### 3.5. The Hypotension Period after Appropriate Antimicrobial Therapy (AAT) Initiation in Subgroup Patients 

Of five leading sources of bacteremia, the highest median (IQR), 178.8 (51.0–342.5) h, of the hypotension period after AAT initiation was observed in 326 patients with pneumonia, followed by 84.9 (42.0–257.5) h in 76 with soft-tissue infections, 75.0 (34.7–217.5) h in 96 with intra-abdominal infections, 48.0 (9.8–105.0) h in 53 with biliary tract infections, and 44.0 (13.0–90.0) h in 171 with urinary tract infections.

Among five major causative pathogens, the highest median (IQR), 105.0 (28.3–317.5) h, of this hypotension period was observed in 124 patients with streptococcal bacteremia, followed by 100.0 (34.1–270.0) h in 51 with *Pseudomonas* bacteremia, 90.0 (39.8–300.0) hours in 121 with staphylococcal bacteremia, 75.5 (24.5–260.0) h in 210 with *Klebsiella* bacteremia, and 72.0 (25.0–240.0) h in 275 with *E. coli* bacteremia. Summarily, regardless of bacteremia sources (Figure 4A) or causative pathogens (Figure 4B), the significantly longer hypotension period after AAT initiation was exhibited in patients within 30-day fatality, compared to that in survivors.

## 4. Discussion

In addition to prompt AAT administration and source control, numerous therapeutic strategies recommended by the SSC guideline for treating patients with severe sepsis and septic shock, such as the fluid challenge technique, appropriate vasoactive medications, and adequate corticosteroid dosing, have been demonstrated to reduce sepsis-related mortality [6]. However, with respect to hemodynamic stability in the response to effective antimicrobial therapy and adverse events of vasopressor agents, no study has measured the period gap between AAT administration and discontinuation of vasopressor agents or attempted to determine the relationship between this time gap and patient outcomes. In the present study, each additional hour of the hypotension period following AAT initiation was significantly associated with an average reduction of 30-day survival rates of 1.1%, after adjusting for independent predictors recognized by the multivariate regression model. 

Whether appropriate empirical antimicrobial therapy has a beneficial effect in patients with bloodstream infections is uncertain; multiple studies have reported no effect [21,22], whereas many have indicated a significant reduction in fatality [4,23,24,25]. However, in cases with bacteremia initially presenting with the critical illness, this discrepancy was diminished. Consistent with past research [3,24] and SSC recommendations [6], each hour of AAT delay resulted in an average increase of 30-day mortality rates of 0.8% herein, after adjusting for confounding factors. Because an linear-by-linear association of the TtAa and the hypotension period following AAT administration and this hypotension period was recognized as the crucial determinant of 30-day fatality herein, we summarily believed that prompt AAT administration might shorten the hypotension period and thereby achieve favourable prognoses. Accordingly, it was reasonable for the implementation of the updated SCC guideline to rapidly administer broad-spectrum antibiotics (within one hour) for reducing the TtAa and apply the crystalloid or fluid resuscitation, adequate source control, and supportive treatment for shortening the hypotension period. 

The severity of comorbidity and bacteremia at onset were powerful indicators of short-term mortality in varied types of bacteremia, such as community-onset bacteremia [3], and bloodstream infections caused by various pathogens, such as Enterobacteriaceae [26], *Staphylococcus* species [27], and *Streptococcus* species [28]. In accord with these reports, fatal comorbidities and the initial presence of the critical illness remained as crucial predictors of unfavorable outcomes in our cohort. Furthermore, similar to numerous ED-based investigations of bloodstream infections [3,29], bacteremia caused by urinary or biliary infections were protective factors against mortality herein.

Our findings are novel but expected. Despite the numerous complex scores and clinical risk factors for mortality prediction using baseline variables captured at bacteremia onset [3,19,28,29], few studies have discussed the relationship between unfavorable outcomes and early clinical failure in the response to initial antimicrobial therapy [30]. Focusing on patients initially experiencing severe sepsis and septic shock, it was reasonably understandable that the timing of discontinuing vasopressor agents (i.e., the hypotension period) was indicative of a substantial parameter in the response to AAT administration herein. Furthermore, the adverse effects of long-term administration of vasopressor agents, such as dysrhythmias and peripheral, renal, mesenteric, or myocardial ischemia, have been comprehensively documented [31]. Such adverse events likely contributed to unfavorable outcomes in our cohort.

This study possesses some limitations. First, in previous studies demonstrating the importance of empirical antimicrobial administration [21,23,24,25], no detailed information following the SSC recommendation was evaluated as covariates. Superior to these reports, to reduce the difference in the quality of care offered by varied medical and nursing staff members, all ED encounters were managed by board-certified physicians and nurses, and all resuscitations were performed using the AHA/ACLS guidelines. Moreover, the impacts of inadequate source control for complicated bacteremia on patient prognoses was concerned here. Second, a few patients with uncertain death dates or incomplete clinical information were excluded from our analysis, contributing to negligible, if any, selection bias. Third, concerning the recall bias, we designed all clinical information to be retrospectively captured by two authors and the two authors inspected the medical records together for any discrepancies. Finally, because all the study hospitals were located in southern Taiwan, the value of increasing mortality risk for prolonging the hypotension period following AAT initiation may not be validated in other communities with variable bacteraemia or comorbidity severity. However, a valuable and novel finding of the present study was that rapider AAT administration was associated with a shorter period of hypotension and the hypotension period after AAT initiation is crucially linked to short-term mortality. Based on our finding, an observational multicentre study designed to prospectively evaluate all bacteremia types should be performed in the future. 

## 5. Conclusions

Irrespective of bacteremia sources and causative microorganisms, the hypotension period after AAT initiation is a crucial determinant of short-term prognoses for patients with community-onset bacteraemia complicated with severe sepsis and septic shock. In such critically ill patients, more rapid AAT administration was associated with the shorter period of hypotension following AAT initiation and thereby achieved more favourable short-term prognoses. In light of these findings to achieve the improved prognoses, we recommend the incorporation of broad-spectrum antimicrobial as empirical therapy into an antibiotic stewardship program and aggressive strategies for reducing the hypotension period.

## Figures and Tables

**Figure 1 jcm-09-02617-f001:**
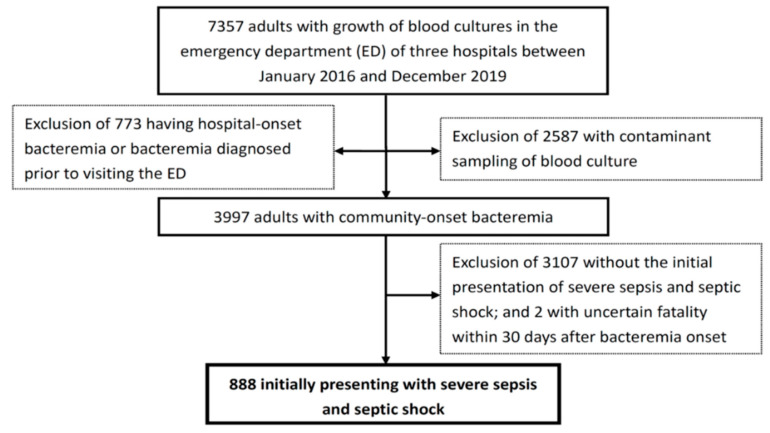
Flowchart of patient selection.

**Figure 2 jcm-09-02617-f002:**
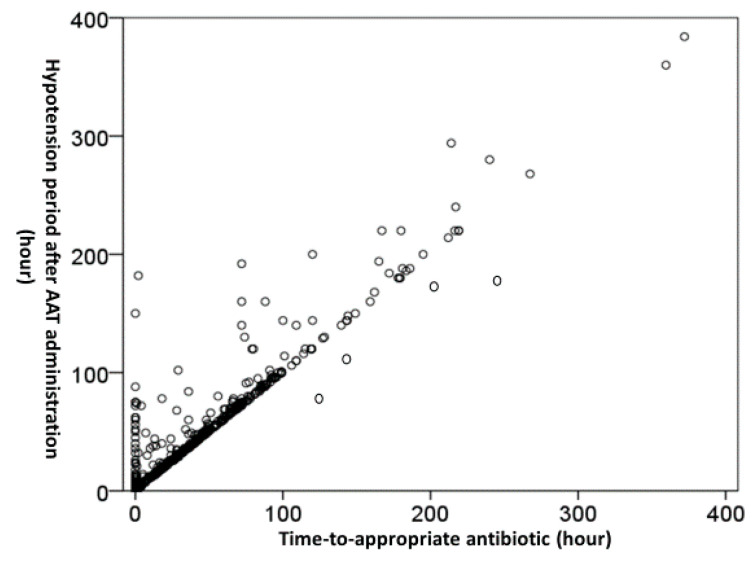
A linear-by-linear association (*γ* = 0.839, *p* < 0.001) of the time-to-appropriate antibiotic and the hypotension period after AAT initiation, tested by *Pearson’s* correlation. AAT = appropriate antimicrobial therapy.

**Figure 3 jcm-09-02617-f003:**
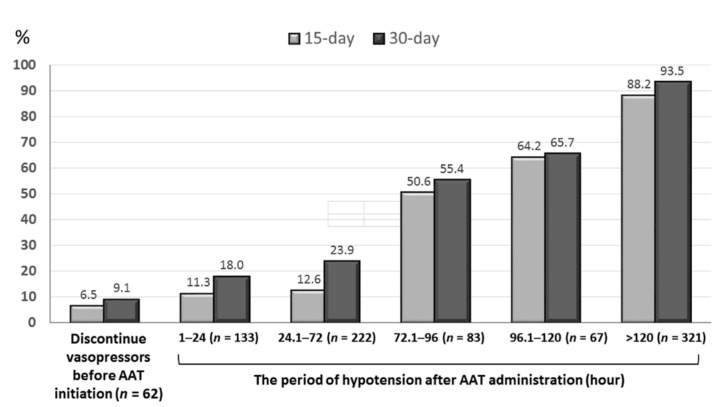
A positive trend of the hypotension period after AAT initiation in the 15-day (*γ* = 0.957, *p* = 0.003) and 30-day (*γ* = 0.975, *p* = 0.001) crude mortality rate, tested by the *Pearson’s* correlation. AAT = appropriate antimicrobial therapy.

**Figure 4 jcm-09-02617-f004:**
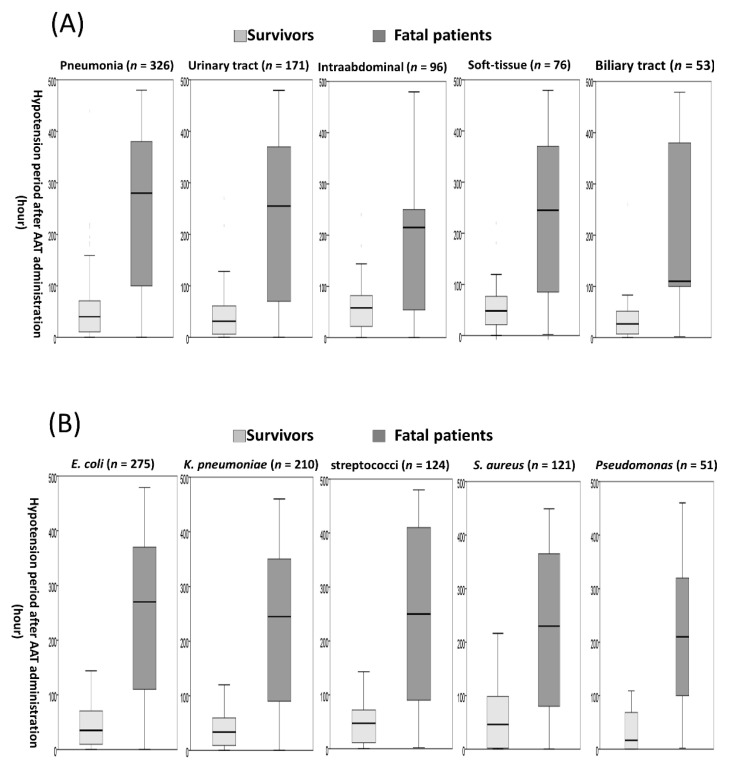
The boxplot of the hypotension period after AAT initiation between survivors and fatal patients within 30 days after bacteremia onset in leading bacteremia sources (**A**) and causative microorganisms (**B**). AAT = appropriate antimicrobial therapy.

**Table 1 jcm-09-02617-t001:** Predictors of 30-day crude mortality in patients having community-onset bacteremia and initially presenting with severe sepsis and septic shock.

Variables	Patient Number (%)		Univariate Analysis		Multivariate Analysis	
	Death, *n* = 472	Survival, *n* = 416	OR (95% CI)	*p* Value	Adjusted OR (95% CI)	*p* Value
Patient demographics						
The elderly, ≥65 years	312 (66.1)	250 (60.1)	1.30 (0.99–1.70)	0.06	NS	NS
Gender, male	272 (57.6)	246 (59.1)	0.94 (0.72–1.23)	0.65	–	–
**Nursing-home residents**	**66 (14.0)**	**34 (8.2)**	**1.83 (1.18–2.83)**	**0.006**	NS	NS
**Time-to-appropriate antibiotic, hours**	**–**	**–**	**–**	**–**	**1.008 (1.006–1.010)**	**<0.001**
**Hypotension period after AAT initiation, hours**	**–**	**–**	**–**	**–**	**1.011 (1.010–1.013)**	**<0.001**
**Inadequate source control**	**26 (5.5)**	**10 (2.4)**	**2.37 (1.13–4.97)**	**0.02**	NS	NS
**Pitt bacteremia score ≥ 4 at onset**	**427 (90.5)**	**245 (58.9)**	**6.63 (4.60–9.53)**	**<0.001**	**3.45 (2.01–5.93)**	**<0.001**
Major bacteremia sources						
**Pneumonia**	**234 (49.6)**	**92 (22.1)**	**3.46 (2.58–4.64)**	**<0.001**	NS	NS
Intra-abdominal	46 (9.7)	50 (12.0)	0.79 (0.52–1.21)	0.28	–	–
**Urinary tracts**	**42 (8.9)**	**129 (31.0)**	**0.22 (0.15–0.32)**	**<0.001**	**0.34 (0.18–0.62)**	**<0.001**
Soft-tissue	40 (8.5)	36 (8.7)	0.98 (0.61–1.57)	0.92	–	–
**Biliary tracts**	**17 (3.6)**	**36 (8.7)**	**0.39 (0.22–0.71)**	**0.002**	**0.20 (0.07–0.58)**	**0.003**
Polymicrobial bacteremia	90 (19.1)	62 (14.9)	1.35 (0.94–1.92)	0.10	–	–
Major causative microorganisms						
***Escherichia coli***	**117 (24.8)**	**158 (38.0)**	**0.54 (0.40–0.72)**	**<0.001**	NS	NS
*Klebsiella pneumoniae*	112 (23.7)	98 (23.6)	1.01 (0.74–1.38)	0.95	–	–
***Streptococcus* species**	**79 (16.7)**	**45 (10.8)**	**1.66 (1.12–2.45)**	**0.01**	NS	NS
***Staphylococcus aureus***	**75 (15.9)**	**46 (11.1)**	**1.52 (1.03–2.25)**	**0.04**	NS	NS
*Pseudomonas* species	33 (7.0)	18 (4.3)	1.66 (0.92–3.00)	0.09	NS	NS
*Enterococcus* species	16 (3.4)	19 (4.6)	0.73 (0.37–1.45)	0.37	–	–
**Fatal comorbidities (McCabe classification)**	**201 (42.6)**	**111 (26.7)**	**2.04 (1.54–2.71)**	**<0.001**	**2.03 (1.27–3.23)**	**0.003**
Major comorbidities						
Hypertension	207 (43.9)	184 (44.2)	0.99 (0.76–1.28)	0.91	–	–
**Malignancies**	**188 (39.8)**	**110 (26.4)**	**1.84 (1.38–2.45)**	**<0.001**	NS	NS
Neurological diseases	169 (35.8)	124 (29.8)	1.31 (0.99–1.74)	0.06	NS	NS
Diabetes mellitus	162 (34.3)	167 (40.1)	0.78 (0.59–1.02)	0.07	0.63 (0.39–1.01)	0.06
Chronic kidney diseases	88 (18.6)	93 (22.4)	0.80 (0.57–1.10)	0.17	–	–
Liver cirrhosis	70 (14.8)	60 (14.4)	1.03 (0.71–1.50)	0.86	–	–
Coronary artery diseases	53 (11.2)	40 (9.6)	1.19 (0.77–1.83)	0.43	–	–

AAT = appropriate antimicrobial therapy; CI = confidence interval; NS = not significant (after processing the backward multivariate regression); OR = odds ratio. Boldface indicates statistical significance with a *p* value of ≤0.05.

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
