# Peer review of "The Hypotension Period after Initiation of Appropriate Antimicrobial Administration Is Crucial for Survival of Bacteremia Patients Initially Experiencing Severe Sepsis and Septic Shock"

_jcm, 2020, doi:10.3390/jcm9082617_

Round 1
Reviewer 1 Report
In this manuscript, the authors have researched and assessed enormous number of case history as to severe sepsis and septic shock in a careful manner. I think this manuscript will be valuable one. However, I am still concerned about some points as an academic article. I describe those points below.
(1) I recommend the authors that the English words will be changed into the easily understandable one.
Example. Page 10, Line No. 211.
Of all the "eligible" patients, major "comorbidities" included hypertension, ..........
(2) Please check the English letters and comma, once more.
Example.
Page 5, Line No. 109. Please change the first large letter.
Page 15, Line No. 288. After the word bacteria, comma and period.
Author Response
n this manuscript, the authors have researched and assessed enormous number of case history as to severe sepsis and septic shock in a careful manner. I think this manuscript will be valuable one. However, I am still concerned about some points as an academic article. I describe those points below.
(1) I recommend the authors that the English words will be changed into the easily understandable one.
Example. Page 10, Line No. 211.
Of all the "eligible" patients, major "comorbidities" included hypertension, ..........
Response: Greatly thanks for your substantial recommendation. The language aspect of the paper has been reviewed for syntactical and grammatical errors by an individual whose mother language is English, and the editorial certification by “Wallace Academic Editing” was attached. For examples, the sentence in line 202 on page 9 was reworded.
(2) Please check the English letters and comma, once more.
Example.
Page 5, Line No. 109. Please change the first large letter.
Page 15, Line No. 288. After the word bacteria, comma and period.
Response: Thanks. These grammatical errors were corrected. Please refer to line 106 on page 5 and line 269 on page 15.

Reviewer 2 Report
Brief summary
The authors report an original article in order to determine the relationship between the hypotension period following administration of appropriate antimicrobials and short-term outcomes, namely on patients having community-onset bacteremia and initially presenting with severe sepsis and septic shock. Several analysis have been included: (i) demographics and clinical characteristics of the study population; (ii) causative microorganisms and susceptibilities; (iii) time-to-appropriate antibiotic, hypotension periods, and mortality rates; (iv) impacts of the time-to-appropriate antibiotic and hypotension period on 30-day mortality and the study of (v) the hypotension period in subgroup patients. The authors conclude that the results indicate that prompt administration of appropriate antimicrobials leads to a shorter period of hypotension and the hypotension period after appropriate therapy initiation is crucially linked to short-term mortality.
Broad comments
- It is a well design study in a relevant topic.
- The manuscript is well organized, containing all the components expected. The article is well-written and is easy to understand.
- The Abstract is concise and adequately summarizes the article content.
- The Introduction can be improved, in order to present an adequate synthesis of the literature. The questions sets out to answer should be clearly exposed.
- The Methods are clearly described and an interesting discussion of the results was considered.
- The limitations of the study are adequately described and discussed in the manuscript (line 339-352). However, a suggestion of specific future studies and direction in this thematic should be stated.
- It is suggested a professional English revision of the manuscript.
- Please revise the sentence “888 bacteremia patients initially presenting with severe sepsis and septic shock (line 202-204)”. There is a lack of accuracy in the way how it is written.
- Enterobacteriaceae should be italicized (line 324).
- Trivial names as “staphylococci”, and “streptococci” should be avoid (lines 325). Please revise the manuscript to the scientific names as Staphylococcus spp. and Streptococcus spp., respectively.
Specific comments
- Please add a full stop at the end of the phrase (line 313).
Reviewer 3 Report
Interesting read, but I have a problem with understanding what is the main message: the hypotension is correlated with mortality/fatality…? I do understand what authors where measuring – but it is not clear what they want to do with the observed outcome? What is a practical usefulness of the observation? The methodology is clear, result are clear but the applicability should be clear– and overall purpose of this MS must be “light motif” in all segments of paper, including Abstract, introduction, discussion, and conclusions. I guess the main “message” is condensed in the sentence:
“the hypotension period after initiation of appropriate antimicrobial therapy is a crucial determinant of short-term prognoses for critically ill patients with community-onset bacteraemia”.
What is operative definition of appropriate antimicrobial therapy? Empirical, based on the outcome?
The period duration of hypotension is one of the prognostic parameters and it is useful since it is independent form patient history. Are there other “operative clinical parameters”, like body temperature, biochemical markers…?
To get adequate prognosis, the relation among multiple prognostic parameters is important. Please specify them and try to make ranking contribution of each.
Author Response
Interesting read, but I have a problem with understanding what is the main message:
Response: Many thanks for your review and questions. In the following paragraph, I had responded to your questions point-by-point.
- The hypotension is correlated with mortality/fatality…?
Response: In our work, the crucial association of 30-day mortality and the hypotension period after AAT administration was proofed by two statistical methods, such as the correlation (Figure 2, page 11) and multivariate regression model (Table 1, page 13-14).
- I do understand what authors where measuring – but it is not clear what they want to do with the observed outcome?
Response: The primary outcome of the present study is the 30-day crude mortality. Please refer to line 101-102 on page 5.
- What is a practical usefulness of the observation?
Response: Focusing on bacteremia adults the critical illness, useful information for clinicians in our work are (i) rapider AAT administration facilitated the shorter period of hypotension (Figure 2, page 11); (ii) the hypotension period after AAT initiation is a crucial determinant of short-term prognoses (Table 1, page 13-14); and (iii) in summary, our finding (Table 1, page 13-14) was equated with the previously establish concept that the rapid administration of appropriate antimicrobials facilitated favorable prognoses is necessary in critical ill patients. Accordingly, to achieve reduced their fatality, we recommended that ED physicians should promptly administer appropriate antibiotics and also adapt the aggressive supportive care to diminish the hypotension period. Please refer to section of conclusion (line 340-345, page 18-19).
- The methodology is clear, results are clear but the applicability should be clear– and overall purpose of this MS must be “light motif” in all segments of paper, including Abstract, introduction, discussion, and conclusions. I guess the main “message” is condensed in the sentence: “the hypotension period after initiation of appropriate antimicrobial therapy is a crucial determinant of short-term prognoses for critically ill patients with community-onset bacteraemia”.
Response: Thanks for your consideration. The message for clinical applicability was inserted in detail in the revised manuscript. First, our hypothesis was inserted and our aims were detailedly revised for more understanding. Please refer to line 59-66 on page 3. Second, we highlighted our finding in the section of discussion. Please refer to the line 287-290 on page 16 and line 294-300 on page 17. Third, the conclusion section was also reworded (line 338-345 on page 18-19).
- What is operative definition of appropriate antimicrobial therapy? Empirical, based on the outcome?
Response: As our previous work (reference 3 and 4), the operative definition of appropriate antimicrobial therapy was listed in line 128-132 on page 6.
6.The period duration of hypotension is one of the prognostic parameters and it is useful since it is independent form patient history. Are there other “operative clinical parameters”, like body temperature, biochemical markers…?
Response: Thanks for your opinion. In addition to antimicrobial therapy and source control, the baseline parameter at bacteremia onset, in terms of patient demographics, bacteremia severity, and comorbidity severity, and other clinical variables, such as causative microorganisms, bacteremia sources, and comorbidity types, were included in univariate and multivariate analyses. In our work, bacteremia severity was assessed by a Pitt bacteremia source, in which the body temperature and other vital signs were calculated. Please refer to line 151-153 on page 7. However, the collection of biochemical markers, such as c-reactive protein, was not routinely examined in all the study hospitals. Because of the nature of this retrospective cohort, this information was not available herein. Because of the severity of bacteremia and comorbidities was enrolled in analyses, the influence of the lacking marker on our finding should be trivial.
- To get adequate prognosis, the relation among multiple prognostic parameters is important. Please specify them and try to make ranking contribution of each.
Response: As the above description, the primary outcome of the present study is the 30-day crude mortality. Generally, the delayed AAT administstaion was evidenced as one crucial determinant of poor patient prognoses in critical ill patients. Similarly, our result also echoes this concept. More importantly, our cohort firstly demonstrated the hypotension period after AAT administration is a crucial determinant of short-term mortality and indicated that rapider administration of appropriate antimicrobials facilitated the shorter period of hypotension. Our meaning was emphasized in the section of conclusion (line 337-345, page 18-19).

Reviewer 4 Report
Title: The hypotension period after initiation of appropriate antimicrobial administration is crucial for the survival of bacteremia patients initially experiencing severe sepsis and septic shock
Authors: Ching-Chi Lee1,2, Chao-Yung Yang3, Chih-Chia Hsieh3, Ming-Yuan Hong3, Chung-Hsun Lee3, Bo-An Su4, Wen-Chien Ko2,5
Revision:
General advice: an extensive language revision by an English mother-language editor is required (suggestions: shorten several periods, avoid excessive use of subordinates adjust syntax).
Abstract and Introduction:
Methods
• Sepsis definition: this is a crucial point to make the work up-to-date with the current literature and medical view. In my opinion, and also considered that this is a retrospective study, Authors should better justify the choice of not adhering to the currently accepted definition of sepsis and septic shock (Sepsis-3, 2016) or, if possible, re-evaluate the data based on this classification.
• Statistics: how the two studied variables, both expressed in hours (i.e. continuous, or discrete variables) were analyzed within a logistic, and not a linear regression model? Please clarify better.
• Statistics: why the two studied variables were not included in univariate analysis? Please clarify
• Statistics: the sentence “Continuous variables were expressed as the means ± standard deviations and compared using Student's t test.” is repeated twice, once referred to normally distributes variables and
once to not normally distributed. Please correct
Line 150: correct lacks
Line 235: correct IRQ to IQR
Figure 3: what does “DC” stand for: please clarify
Figure 3: correct appropaite
Figure 2 and 4: the word “Periods” is not clear. Explain better what does it stand for, or replace with “hours” if appropriate
Discussion
Line 305: in the sentence invert “time between initiation…and discontinuation”. It seems more logical to follow the chronological order of events
Line 308-309: “a positive linear-by-linear association was noticed between the hypotension period and the 30-day mortality rate.” Where is this data? In the results section the only linear-by-linear regression is between the two studied variables (figure 2). Please clarify and/or correct
The statement “However, a valuable and novel finding of the present study is that prompt administration of appropriate antimicrobials leads to a shorter period of hypotension and the hypotension period after appropriate therapy initiation is crucially linked to short-term mortality” is not entirely correct: TtAa and the resolution of hypotension are correlated, as showed in figure 2, but the causative correlation is not demonstrated. Please correct this concept, here and elsewhere.
Author Response
Revision:
General advice: an extensive language revision by an English mother-language editor is required (suggestions: shorten several periods, avoid excessive use of subordinates adjust syntax).
Response: Many thanks for your review and questions. Using twol abbreviations, such as AAT (appropriate antimicrobial therapy) and TtAa (time-to-appropriate antibiotic), we had avoided the excessive use of subordinates adjust syntax. Also, the language aspect of the paper has been reviewed for syntactical and grammatical errors by an individual whose mother language is English, and the editorial certification by “Wallace Academic Editing” was attached.
Abstract and Introduction:
Methods
• Sepsis definition: this is a crucial point to make the work up-to-date with the current literature and medical view. In my opinion, and also considered that this is a retrospective study, Authors should better justify the choice of not adhering to the currently accepted definition of sepsis and septic shock (Sepsis-3, 2016) or, if possible, re-evaluate the data based on this classification.
Response: Thanks for your opinion. We agree the Sepsis-3 guideline proposed the quick Sequential Organ Failure Assessment (qSOFA) as a replacement for the systemic inflammatory response syndrome (SIRS) criteria. However, this revision remains debated in the literature, such as the opposite comments indicated by Dr. Simpson (DOI: http://dx.doi.org/10.1016/j.chest.2016.02.653), Professor Vincent (DOI: 10.1186/s13054-016-1389-z), and Dr. Giamarellos-Bourboulis (DOI: http://dx.doi.org/10.1016/j.cmi.2016.11.003). Accordingly, the traditional definition of severe sepsis and septic shock was adapted herein.
Statistics: how the two studied variables, both expressed in hours (i.e. continuous, or discrete variables) were analyzed within a logistic, and not a linear regression model? Please clarify better.
Response: Thanks for your questions. The logistic and linear regression model are both popular multivariate regression models. But the leading difference between theses was the character of the output (dependent variable). For the logistic regression model, the suitable output is a probability ranging from 0 (not going to happen) to 1 (definitely will happen), or a categorization that says something is either part of the category or not part of the category. But the dependent variable in the linear regression model is continuous. A continuous value can take any value within a specified interval (range) of values.
In our work, the dependent variable was 30-day crude mortality. Accordingly, it was reasonable that we choose the logistic regression model as multivariate analyses.
Statistics: why the two studied variables were not included in univariate analysis? Please clarify
Response: Thanks for your opinions. To avoid the reader's confusion, the statement in the statistic section was revised. Please refer to line 167-170 on page 7-8.
- Statistics: the sentence “Continuous variables were expressed as the means ± standard deviations and compared using Student's t test.” is repeated twice, once referred to normally distributes variables and once to not normally distributed. Please correct
Response: Thanks for your opinions. The section of statistic was reconstructed. Please refer to line 162-166 on page 7.
Line 150: correct lacks
Response: The grammatical error was reworded. Please refer to line 145 on page 7.
Line 235: correct IRQ to IQR
Response: The error was reworded. Please refer to line 225 on page 10.
Figure 3: what does “DC” stand for: please clarify
Figure 3: correct "appropaite"
Response: The Figure 3 was revised. Please refer to page 11
Figure 2 and 4: the word “Periods” is not clear. Explain better what does it stand for, or replace with “hours” if appropriate
Response: The word in the y-axis was revised in Figure 2 (page 11) and Figure 4 (page 15-16).
Discussion
Line 305: in the sentence invert “time between initiation…and discontinuation”. It seems more logical to follow the chronological order of events
Response: This sentence was reworded. Please refer line 285-286 on page 17.
Line 308-309: “a positive linear-by-linear association was noticed between the hypotension period and the 30-day mortality rate.” Where is this data? In the results section the only linear-by-linear regression is between the two studied variables (figure 2). Please clarify and/or correct
Response: Thanks for your consideration. The figure 2 indicated the linear-by-linear association between the hypotension period after AAT administration and the 15-day or 30-day crude mortality rate. In further analyses shown in Table 1, additional hour of the hypotension period following AAT initiation was significantly associated with an average reduction of 30-day survival rates of 1.1%, after adjusting for independent predictors recognized by the multivariate regression model. To avoid the reader's confusion, the original sentence was revised to line 296-300 on page 17.
The statement “However, a valuable and novel finding of the present study is that prompt administration of appropriate antimicrobials leads to a shorter period of hypotension and the hypotension period after appropriate therapy initiation is crucially linked to short-term mortality” is not entirely correct: TtAa and the resolution of hypotension are correlated, as showed in figure 2, but the causative correlation is not demonstrated. Please correct this concept, here and elsewhere.
Response: Thanks for your consideration. This and the related sentence were reworded throughout the revised manuscript. Please refer to line 321-323 and line 340-342 on page 18.

Round 2
Reviewer 2 Report
The authors have address the reviewer comments.
Author Response
Many thanks again for your substantial review.
Reviewer 3 Report
Thank you, now I get the clear picture that the most significant finding is the fact that if hypotension continues after AAT, the patient is in trouble (how long?). Thank you. Is it in the causal relation to AAT or is it just too late...or a combination? Otherwise, I would publish this without delay.
Author Response
Thank you, now I get the clear picture that the most significant finding is the fact that if hypotension continues after AAT, the patient is in trouble (how long?). Thank you.
Response: Many thanks again for your substantial review. As shown in Figure 3, a positive trend of the hypotension period after AAT initiation in the 15-day or 30-day crude mortality rate was disclosed. in other words, as the hypotension period after AAT initiation increased, the short-term survival dropped.
Is it in the causal relation to AAT or is it just too late...or a combination? Otherwise, I would publish this without delay.
Response: Greatly thanks for your question. This a good question for clinicians. In my opinions, antimicrobial-resistant microorganisms and the lacking of treatment guidelines were the leading causes linked to inappropriate administration of empirical antimicrobials. Therefore, in the literature, there were numerous reports dealing with the reduced mistake of empirical administstaion by adapting the antimicrobial stewardship program (Clinical Infectious Diseases 2006:42(suppl 2): S90–S95, Clinical Infectious Diseases 2010;51(9):1074–1080, and Clinical Microbiology Review 2005;18(4):638–656). So, risk factors of inappropriate administration of empirical antibiotics is necessary to disclose in our manuscript and is only offered inside the study hospitals for the increased quality of patient care.
